# Assessment of Postvaccination Neutralizing Antibodies Response against SARS-CoV-2 in Cancer Patients under Treatment with Targeted Agents

**DOI:** 10.3390/vaccines10091474

**Published:** 2022-09-05

**Authors:** Flora Zagouri, Alkistis Papatheodoridi, Michalis Liontos, Alexandros Briasoulis, Aimilia D. Sklirou, Efthymia Skafida, Oraianthi Fiste, Christos Markellos, Angeliki Andrikopoulou, Konstantinos Koutsoukos, Maria Kaparelou, Eirini Gkogkou, Ioannis P. Trougakos, Meletios-Athanasios Dimopoulos, Evangelos Terpos

**Affiliations:** 1Department of Clinical Therapeutics, National and Kapodistrian University of Athens, Alexandra Hospital, 11528 Athens, Greece; 2Section of Cell Biology and Biophysics, Department of Biology, School of Sciences, National and Kapodistrian University of Athens, Panepistimiopolis, 15701 Athens, Greece

**Keywords:** SARS-CoV-2, vaccination, cancer, ARTA, CDK4/6 inhibitors, PARP inhibitors

## Abstract

The administration of a third dose of a vaccine against SARS-CoV-2 has increased protection against disease transmission and severity. However, the kinetics of neutralizing antibodies against the virus has been poorly studied in cancer patients under targeted therapies. Baseline characteristics and levels of neutralizing antibodies at specific timepoints after vaccination were compared between patients suffering from breast, ovarian or prostate cancer and healthy individuals. Breast cancer patients were treated with cyclin D kinase 4/6 inhibitors and hormonal therapy, ovarian cancer patients were treated with poly (ADP-ribose) polymerase inhibitors and prostate cancer patients were treated with an androgen receptor targeted agent. Levels of neutralizing antibodies were significantly lower in cancer patients compared to healthy individuals at all timepoints. Antibodies’ titers declined over time in both groups but remained above protective levels (>50%) at 6 months after the administration of the second dose. The administration of a third dose increased neutralizing antibodies’ levels in both groups. The titers of protective against SARS-CoV-2 antibodies wane over time and increase after a third dose in cancer patients under treatment.

## 1. Introduction

Almost three years ago, the outbreak of the COVID-19 pandemic led to an unprecedented public health emergency worldwide. Vaccination with any of the available approved vaccines against SARS-CοV-2 managed to reduce new infections and disease severity. However, the declining efficacy of the vaccines over time and the development of new viral variants resulted in COVID-19 cases among vaccinated individuals [1,2,3]. Administration of a booster 3rd dose increased protection against the virus and was performed in several countries including Greece [4]. Cancer is considered a strong risk factor for severe disease and death from SARS-CoV-2 [5], thus, cancer patients were prioritized to receive a 3rd dose of the COVID-19 vaccine. However, the safety and efficacy of these vaccines in cancer patients undergoing treatment with targeted therapies are not yet clarified. In the present prospective study (NCT047443388), we attempted to evaluate the immune response of either patients with solid tumors or healthy individuals after 3 and 6 months from the 2nd dose of a COVID-19 vaccine, as well as 1 month after a booster 3rd dose. More specifically, we report the levels of neutralizing antibodies (NAb) against SARS-CoV-2 in patients with prostate, breast cancer and ovarian cancer under treatment in comparison to healthy controls.

## 2. Materials and Methods

### 2.1. Study Design

This is a prospective observational two-cohort study aiming at assessing the kinetics of antibodies against SARS-CoV-2 in healthy volunteers and cancer patients receiving the vaccination against SARS-CoV-2. Cohort 1 enrolled volunteers, healthy or with chronic non-neoplastic diseases. Cohort 2 enrolled patients with hematological malignancies or solid tumors in various phases of their treatment. The study was approved by the respective Ethical Committees in accordance with the Declaration of Helsinki and the International Conference on Harmonization for Good Clinical Practice. All patients and controls provided written informed consent prior to enrollment in the study. The confidentiality of the participants’ data was maintained in accordance with the rules of the General Data Protection Regulation (GDPR). All of the subject identities were kept strictly private following the principles of ‘pseudonymisation’. All patients were recruited between January and May 2021 and were then followed up to 18 months, unless death or change of treatment had occurred.

### 2.2. Patients

Volunteers (Cohort 1), as well as patients with histologically confirmed prostate, ovarian or breast cancer from Cohort 2 of our study, were included in this analysis. All patients with prostate cancer were under treatment with an androgen receptor targeted agent (ARTA), all patients with ovarian cancer were treated with poly (ADP-ribose) polymerase (PARP) inhibitors and all patients with breast cancer were treated with cyclin D kinase 4/6 (CDK4/6) inhibitors and hormonal therapy. Other inclusion criteria for the study included age above 18 years old and eligibility for vaccination. Healthy individuals included in another cohort of the study were used as controls. Major exclusion criteria for both patient- and control- cohorts included the presence of: (i) another active malignant disease; (ii) autoimmune disease; (iii) Human Immunodeficiency Virus (HIV) and/or active hepatitis B and C infection; and (iv) prior diagnosis of COVID-19 infection using polymerase chain reaction (PCR) test. Patients’ and controls’ baseline characteristics including age, sex, body mass index (BMI) and significant comorbidities were collected prior to 1st dose vaccination (baseline). Adverse events due to vaccination were collected from the medical files of the patients.

### 2.3. Neutralizing Antibodies Detection

Serum samples were obtained from patients and controls at the Department of Clinical Therapeutics in Alexandra General Hospital of Athens, on several timepoints regarding their vaccination status for SARS-CoV-2 and stored at −80 °C. More specifically, blood collection was performed prior to 1st dose (baseline), prion to 2nd dose, 1 month after the 2nd dose, 3 months after the 2nd dose, 6 months after the 2nd dose and 1 month after the 3rd dose. Serum was separated within four hours of blood collection and stored at −80 °C until the day of measurement. The FDA-approved ELISA, cPass™ SARS-CoV-2 NAb Detection Kit (GenScript, Piscataway, NJ, USA) was used for the detection of NAb. The kit detects neutralizing antibodies that interact and block the connection between the receptor binding domain of the viral spike glycoprotein and the ACE2 host cell surface receptor. The interaction is highly specific and in contrast to the conventional virus and pseudovirus neutralizing assays it is faster and there is no need to be performed in specialized biosafety level 2 or 3 facilities. Seropositivity was considered as a result of ≥30% of signal inhibition, while a Nab titer of at least 50% of signal inhibition was associated with clinically relevant viral inhibition. Samples from both patients and controls were measured in each ELISA plate.

### 2.4. Statistical Analysis

Baseline demographics, co-morbidities, and the NAb levels were compared between the 2 groups, Chi-square test for categorical variables and unpaired t-test or Wilcoxon signed-rank test (as appropriate) for continuous variables. Mixed models were performed using direct likelihood estimation with fixed effects of antibody titers, the timing of measurement, and interaction of antibody titers by the timing of measurement. An unstructured covariance matrix was used to model within-patient error. All data extraction and statistical analyses were conducted using Stata Version 17.0 (College Station, Texas). All significance tests were two-tailed and conducted at the 5% significance level.

## 3. Results

### 3.1. Patients Characteristics

In total 102 cancer patients with a median age of 67.0 years (Interquartile Range 57.0–74.0 years) and 275 healthy individuals with a median age of 54.0 years were included in the study. The baseline characteristics of the patients are described in Table 1. Twenty-eight of the 102 cancer patients (27.5%) were diagnosed with metastatic prostate cancer and were treated with Androgen Receptor Targeted Agents (ARTAs), 30 patients (29.4%) with metastatic breast cancer under treatment with CDK4/6 inhibitors and 44 (43.1%) with advanced epithelial ovarian cancer that received PARP inhibitors. Most patients (91.1%) received mRNA vaccines, namely BNT162b2 and mRNA-1273 and only 8.9% of patients received the AZD1222 vaccine. The most common comorbidities in the patients’ population included cardiovascular disease in 32 patients, diabetes mellitus in 9 patients and dyslipidemia in 16 patients.

### 3.2. Kinetics of Neutralizing Antibodies against SARS-CoV-2 Postvaccination

NAb kinetics among patients and controls within the observation period are presented in Table 2. More specifically, on baseline assessment on Day 1, prior to 1st dose vaccination, median NAb levels did not differ between patients and controls (17.5% [11.5–23.4] vs. 14.2% [8–22.3], *p* = 0.1). After the 1st dose, antibody levels were increased in both patients and controls, however, patients presented significantly lower levels than the healthy individuals (27.3% [13–32.3] vs. 54.2%, [39.5–69.8], *p* < 0.001). One month after the 2nd dose both patients and controls developed clinically relevant viral inhibition (NAb levels > 50%); however, once again healthy controls presented with higher titers of antibodies than cancer patients (96.4% [94–97] vs. 88.3%, [53.4–95.2], *p* < 0.001). During 3-month and 6-month follow-ups, NAb titers seem to decline in both groups as shown in Figure 1. Still both groups present with protective levels above 50%, and cancer patients had lower levels than controls (at 3 months; 70.5% [31.3–90.7] vs. 92.3% [84.4–96] *p* < 0.001 and at 6 months 55.9%, [32.2–67.1] vs. 80.8% [63.4–89.7], *p* < 0.001, respectively).

A 3rd booster dose was administered in 11 cancer patients and 150 healthy individuals. Administration of the 3rd dose managed to increase NAb levels in both groups (94% [78.2–97.1] in patients compared to 97.6% [97–97.9] in controls, *p* < 0.001 (Figure 1).

No safety issues were identified by administration of any dose irrespectively of cancer treatment. The most common adverse events following vaccination were fever, site injection reactions and fatigue. In total, 38 patients (37.25%) developed any adverse event during their vaccination courses, but none of them was significant and all were self-resolved.

## 4. Discussion

The COVID-19 pandemic has strongly affected cancer patients. It is known that the diagnosis of cancer is related to increased mortality from SARS-CoV-2 infection. In parallel, delays in cancer treatment and follow-ups due to restrictions imposed during the COVID-19 pandemic negatively affected survival and increased the risk of late diagnosis of the disease and its progression [6]. Therefore, a vaccination schedule that successfully protects cancer patients was of great importance. The implementation of vaccination programs has proved to have protective effects on the total population [7,8,9] but the immunological response among cancer patients was not well documented. This is specifically true for specific populations of cancer patients that receive targeted therapies. We—among others—have recently reported the kinetics of NAb against SARS-CoV-2 in cancer patients up to 1 month after the second vaccination dose of either the BNT162b2, or the mRNA-1273, or the AZD1222 vaccines. More specifically, we have analyzed patients with prostate cancer under treatment with an androgen receptor targeted agent [10], ovarian cancer receiving treatment with PARP inhibitors [11] and breast cancer patients under treatment with CDK4/6 inhibitors [12].

The emergence of the Delta and Omicron variants though has clearly shown that immunological responses are short-lived and booster vaccination is required [13]. Under this perspective we present in the current study, the kinetics of NAb in all three abovementioned cohorts in a long-term follow-up of 3 and 6 months as well as 1 month after administration of a 3rd booster dose, compared to healthy individuals. To our knowledge, this is the first study assessing the natural course of the antibodies in patients with solid tumors under specific targeted treatment.

Our results demonstrate that cancer patients under targeted therapy with either CDK4/6, PARP inhibitors or ARTA have lower levels of NAb after vaccination compared to healthy controls at all timepoints. Several studies conducted over the last three years suggest that these specific targeted treatments can be safely administered during the COVID pandemic [14], while PARP inhibitors and ARTA may have a protective effect against severe infection [15,16]. However, our results are consistent with our previous reports that although cancer patients have protective antibody levels after the 2nd dose of the vaccine, their titers are lower than those of the healthy population [10,11,12]. This could be attributed primarily to the disease itself that constitutes a state of immunodeficiency related to the attenuated immune response to SARS-CoV-2 vaccination reported here. In addition though, specific characteristics of the targeted therapies studies may also contribute to this result. Neutropenia and lymphopenia are common adverse events of CDK4/6 inhibitors and this could lead to decreased production of neutralizing antibodies against SARS-CoV-2 after vaccination [12,17]. In addition, PARP inhibitors are implicated in immune modulation and have shown that they can negatively affect the maturation and antigen-presenting function of dendritic cells [18]. This is consistent with our results regarding immunological response to SARS-CoV-2 vaccination in ovarian cancer patients under treatment with PARP inhibitors.

As has been described before, NAb titers increase after the 2nd dose of the vaccine but tend to decline over time in both groups [1,19,20]. More specifically, 1 month after the administration of the 2nd of dose of the vaccine, NAb levels reach a peak in both patients and controls, as depicted in Figure 1. Over time, though antibody titers gradually wane, they remain above clinically relevant protective levels (>50%) even at 6 months after the 2nd dose. However, administration of a 3rd dose leads to a booster increase in the antibodies’ levels in patients and controls. It is of interest that even after the administration of the 3rd dose the antibodies’ levels were significantly lower in cancer patients compared to healthy controls.

All three doses of the vaccines BNT162b2, mRNA-1273, and AZD1222 were safely administered and well tolerated by patients.

Our study is limited by the relatively small number of patients included, since it is difficult to identify all factors associated with the vaccines’ effectiveness and the disease prognosis. In this cohort of the study, we have analyzed patients with different neoplasms receiving targeting agents and this could introduce bias in our analysis. All these patients though received chemotherapy-free regimens commonly prescribed in cancer patients. Moreover, the patients received either mRNA or viral vector vaccines, but the clinical efficacy of all three vaccines is similar and only mRNA vaccines were used for the booster dose in Greece. Therefore, despite limitations, our results are in accordance with our previous publication and current global guidelines suggesting that cancer patients should be offered a 3rd dose of vaccination against COVID-19.

Nevertheless, since novel variants of the virus emerge, the maintenance of precautions against transmission remains important, especially for vulnerable patients such as cancer patients. At the same time, the concept of multiple booster doses remains of interest. Several studies have already examined the effectiveness of a fourth dose of the vaccine, suggesting that a second booster dose is protective against severe diseases in adults older than 60 years of age [21,22]. Even though the role of this dose in the setting of cancer patients is not yet clarified, international health agencies and guidelines propose the administration of a second booster dose in immunocompromised patients, including cancer patients under treatment.

## 5. Conclusions

In conclusion, while clinically relevant viral inhibition of SARS-CoV-2 infection may wane over time after administration of the 2nd dose of COVID-19 vaccines both in cancer patients and in healthy controls, a booster 3rd dose leads to an increase in the antibodies’ levels in both groups.

## Figures and Tables

**Figure 1 vaccines-10-01474-f001:**
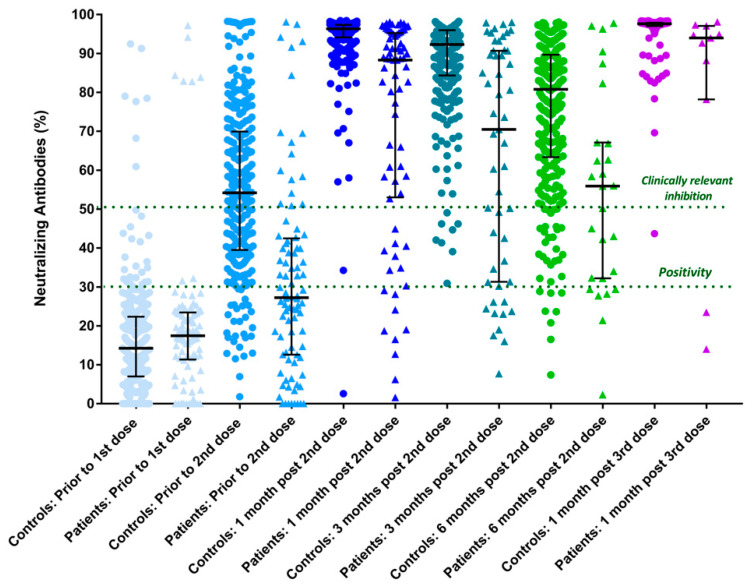
Kinetics of neutralizing antibodies (NAb) in cancer patients receiving targeted therapies and matched controls, at several timepoints after COVID-19 vaccination.

**Table 1 vaccines-10-01474-t001:** Baseline characteristics of the patients.

Variables	Total Population; Median (IQR)
**Age**	67.0 (57.0–74.0)
**Sex**	
Male	28 (27.45%)
Female	74 (72.55%)
**Type of cancer**	
Prostate cancer	28 (27.45%)
Breast cancer	31 (30.39%)
Ovarian cancer	43 (42.15%)
**Type of therapy**	
ARTA	28 (27.45%)
CDK4/6i	31 (30.39%)
PARPi	43 (42.15%)
**Comorbidities**	
Yes	71 (69.61%)
None	26 (25.49%)
Missing	5 (4.90%)
**Type of vaccine**	
BNT162b2	81 (79.41%)
AZD1222	9 (8.90%)
mRNA-1273	12 (11.74%)
**Vaccine-related adverse events**	
Pain at injection site	20 (19.60%)
Fatigue	17 (20%)
Fever	7 (8.24%)
None	36 (42.35%)

**Table 2 vaccines-10-01474-t002:** Temporal trends of neutralizing antibody titers in patients and controls.

Time	Mean NAb Titers ± SD		*p* Value	Median NAb Titers, IQR		*p* Value
	Patients	Controls		Patients	Controls	
Baseline before 1st dose	20.4 ± 20.3 (*n* = 88)	17.4 ± 16 (*n* = 275)	0.049	17.5, 11.5–23.4	14.2, 8–22.3	0.1
Prior to 2nd dose	31.5 ± 24.6 (*n* = 88)	49.2 ± 24.3 (*n* = 269)	<0.001	27.3, 13–32.3	54.2, 39.5–69.8	<0.001
1 month after 2nd dose	72.2 ± 28.5 (*n* = 72)	89.7 ± 17 (*n* = 291)	<0.001	88.3, 53.4–95.2	96.4, 94–97	<0.001
3 months after 2nd dose	63.4 ± 29.4 (*n* = 51)	84.3 ± 18 (*n* = 288)	<0.001	70.5, 31.3–90.7	92.3, 84.4–96	<0.001
6 months after 2nd dose	54 ± 25.8 (*n* = 27)	72.9 ± 20.6 (*n* = 278)	<0.001	55.9, 32.2–67.1	80.8, 63.4–89.7	<0.001
1 month after 3rd dose	79.3 ± 30.5 (*n* = 11)	94.8 ± 10 (*n* = 150)	<0.001	94, 78.2–97.1	97.6, 97–97.9	<0.001

## Data Availability

All data generated or analysed during this study are included in this published article.

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
