# Peer review of "Assessment of Postvaccination Neutralizing Antibodies Response against SARS-CoV-2 in Cancer Patients under Treatment with Targeted Agents"

_vaccines, 2022, doi:10.3390/vaccines10091474_

Round 1

Reviewer 1 Report

In the manuscript by Zagouri et al, the authors have characterized the neutralizing antibody titers against SARS-CoV-2 in cancer patients who have been under distinct treatments. It is a strength that authors followed these patients and healthy donors for several months, for example, some donors were followed until 1 month after the 3rd dose. It is an interesting finding that the antibody responses in patients were generally weaker than those in healthy individuals. However, the current manuscript suffers from some flaws.

1. The authors stated in the manuscript that neutralizing antibody titers were measured by using an FDA-approved ELISA kit. I think more explanation of this kit needs to be provided to explain how an ELISA kit could measure the titer of neutralizing antibodies (nAb)? Actually, I do not think there is any ELISA kit could be used to precisely measure the nAb titer. I think an in vitro pseudovirus neutralizing assay should be more precise and should be used in this study.

2. Nowadays, many SARS-CoV-2 variants have appeared, while the original wild-type SARS-CoV-2 have been vanished. However, in this study, only antibody activities against wild-type SARS-CoV-2 were measured. I think it is more important to measure the antibody activity against new Omicron variants, such as BA.4/5.

3. Only 11 cancer patients were administered the 3rd booster dose. This number seems a little bit low compared with the number of healthy individuals who have received the 3rd dose. Is there any possibility to recruit more donors?

4. The authors mainly compared the cancer patients and the healthy donors, but did the author try to compared those patients under distinct treatments? However, no such comparison was included in the manuscript. Is it because the patient numbers of different groups are too low?

5. I suggest a slight modification in Figure 1 by adding those individual dots instead of only showing the median Ab titers.

Reviewer 2 Report

In this study, the authors studied the kinetics of neutralizing antibodies against SARS-Cov- in cancer patients under targeted therapies. Levels of neutralizing antibodies were significantly lower in cancer patients compared to healthy individuals at all timepoints. Antibodies’ titers declined over time in both groups but remained above protective levels (>50%) at 6 months after the administration of the second dose. Administration of a third dose increased neutralizing antibodies’ levels in both groups. The titers of protective against SARS-CoV-2 antibodies wane over time and increase after a third dose in cancer patients under treatment. The study design and presentation are detailed. The data presented are generally strong, and appear convincing.

Major comments:

1.     As mentioned in the manuscript, cancer patients under targeted therapy with either CDK4/6, or PARP inhibitors or ARTA have lower levels of NAb after vaccination compared to healthy controls at all timepoints. The authors should discuss the explanation of it in the discussion part.

Minor comments:

1.     All number of reference citation should be before “.”.

2.     Line39, 43 and 47, should be “COVID-19”.

3.     Line 48-49, “we report the levels of neutralizing antibodies (NAb) against SARS COV2 in patients” should be “SARS-CoV-2”.

4.     Line74, “and stored in -80oC.” The wrong spelling of the temperature unit.

5.     Line123, “p<0.001” should be “P<0.001”.

Reviewer 3 Report

This kind of study have interest to the community in the face of the need to have enough information about the response of vaccines, and the need of more doses or boosters to protect people, specifically the vulnerable people. So, I congratulate to the authors for this work .

 The study’s objectives are assessing the response for SARS-CoV-2 vaccines in cancer patients. 

According with the author thekey findings of the study are that the SARS-CoV-2 antibodies wane over time and increase after third dose in cancer patients and healthy individuals.

The study demonstrates technical rigor. Ill just comment on some aspects of this paper with the intention that these comments will help you to improve it and so it may be easier to read. 

Major comments 

Matherial and methods section

·       I miss a subsection to explain the design of this study. In this subsection, the authors should describe the setting, locations, periods of recruitment and follow-up.

·       I believe that the authors should be better describe the characteristics of the group control and they must explain how the matching was done.

·       There is no mention to potential biases, for exemple: the time of treatment can influence the response of the vaccine?, the different characteristics of the groups can influence the antibodies kinetics? The type of vaccine could influence the results?....

Resullts section

·       The authors describe the patients group but not the control group. Are they statistically significant differences?

·       You present safety data in the results and this is not described in matherial and methods. You should explain how do you collect this data.

Discussion section:

You explaine that the vaccines are well tolerated for patients and controls and BMI an co-morbidities did not seem to affect the kinetic of Nab. This variables don’t have described in the matherial and methods and neither in the results. This requires to include the variables description and the statistically treatment to reach this conclusion.

I agree with the authors that the number of patients included are small and it might not be enough to support their conclusion.

 Minor comments 

I don’t understand why active hepatitis B or C infection are major exclusion criteria and no others infections.

In the subsection of neutralizing antibodies detection you describe” that seropositivity was considered as a result of >30%”. My question is >30% of what?

Round 2

Reviewer 1 Report

The authors have responded productively to the reviewer's concerns. I recommend publication.

Reviewer 3 Report

I congratulate the authors. The corrections done have improved the understanding of the paper.